# Echinacea: Bioactive Compounds and Agronomy

**DOI:** 10.3390/plants13091235

**Published:** 2024-04-29

**Authors:** Fatemeh Ahmadi, Khalil Kariman, Milad Mousavi, Zed Rengel

**Affiliations:** 1UWA School of Agriculture and Environment, The University of Western Australia, Perth, WA 6009, Australia; khalil.kariman@uwa.edu.au (K.K.); milad.mousavi@uwa.edu.au (M.M.); zed.rengel@uwa.edu.au (Z.R.); 2Institute for Adriatic Crops and Karst Reclamation, 21000 Split, Croatia

**Keywords:** abiotic stress, hydroponics, medicinal plants, secondary metabolites

## Abstract

For centuries, medicinal plants have been used as sources of remedies and treatments for various disorders and diseases. Recently, there has been renewed interest in these plants due to their potential pharmaceutical properties, offering natural alternatives to synthetic drugs. Echinacea, among the world’s most important medicinal plants, possesses immunological, antibacterial, antifungal, and antiviral properties. Nevertheless, there is a notable lack of thorough information regarding the echinacea species, underscoring the vital need for a comprehensive review paper to consolidate existing knowledge. The current review provides a thorough analysis of the existing knowledge on recent advances in understanding the physiology, secondary metabolites, agronomy, and ecology of echinacea plants, focusing on *E. purpurea*, *E. angustifolia*, and *E. pallida*. Pharmacologically advantageous effects of echinacea species on human health, particularly distinguished for its ability to safeguard the nervous system and combat cancer, are discussed. We also highlight challenges in echinacea research and provide insights into diverse approaches to boost the biosynthesis of secondary metabolites of interest in echinacea plants and optimize their large-scale farming. Various academic databases were employed to carry out an extensive literature review of publications from 2001 to 2024. The medicinal properties of echinacea plants are attributed to diverse classes of compounds, including caffeic acid derivatives (CADs), chicoric acid, echinacoside, chlorogenic acid, cynarine, phenolic and flavonoid compounds, polysaccharides, and alkylamides. Numerous critical issues have emerged, including the identification of active metabolites with limited bioavailability, the elucidation of specific molecular signaling pathways or targets linked to echinacoside effects, and the scarcity of robust clinical trials. This raises the overarching question of whether scientific inquiry can effectively contribute to harnessing the potential of natural compounds. A systematic review and analysis are essential to furnish insights and lay the groundwork for future research endeavors focused on the echinacea natural products.

## 1. Introduction

Throughout history, humans have employed various strategies to treat a diverse range of diseases and health disorders, including using medicinal plants [1]. As an herbal remedy deeply rooted in traditional medicine, echinacea has garnered attention from researchers, health enthusiasts, and individuals seeking natural wellness solutions [2]. The genus *Echinacea*, known as coneflower, comprises ten species, all originally native to North America [3]. To meet the growing demand for its raw materials, echinacea cultivation has expanded beyond its native North American habitats and now spans nearly every continent [4].

The classification of echinacea species is based on morphological and anatomical features [5]. There are ten well-known echinacea species (The International Plant Names Index and World Checklist of Vascular Plants, 2023); three of them, namely *E. purpurea*, *E. pallida*, and *E. angustifolia*, have significant biological and pharmaceutical importance due to the extensive use of their roots and aerial parts in herbal medicines and dietary supplements [6,7,8]. Herbal products from these plants are top sellers in North America and Europe, generating more than USD 300 million annually in the U.S. alone [9].

*Echinacea purpurea* (L.) Moench is one of the most significant medicinal plants worldwide among the echinacea species, offering more than 20 documented health benefits [10]. Extracts from *E. purpurea* are well-known for their purported capability to boost the immune system. They are frequently used as dietary supplements to enhance immune responses and reduce symptoms of common colds and respiratory issues, such as COVID-19 [9]. Numerous research studies have delved into the effectiveness of *E. purpurea* extracts by shortening the length and lessening the severity of symptoms caused by colds [11,12].

Numerous in vivo studies investigating the immunomodulatory and anti-inflammatory properties of *E. purpurea* have indicated that its use boosts innate immunity and fortifies the immune system defenses against pathogenic infections [7,11,12]. This is achieved through the activation of key immune system components such as neutrophils, macrophages, polymorphonuclear leukocytes, and natural killer cells [13]. *Echinacea purpurea* is renowned for its rich phytochemical content, enhancing its medicinal value [7]. It contains a variety of compounds, such as phenolic acids, alkylamides, polysaccharides, flavonoids, and essential oils, that are thought to support its immunomodulatory, anti-inflammatory, and antioxidant activities [14]. The roots, leaves, and flowers are used in herbal supplements, with the root complex mixture of alkamides, ketoalkenes, CADs, polysaccharides, and glycoproteins believed to boost their immune and anti-inflammatory effects [15]. The composition and concentration of these bioactive phytochemicals can vary among different echinacea species (Table 1). For instance, *E. purpurea* roots are rich in echinacoside, which is rarely found in the roots of other echinacea species and exhibit notably high levels of cynarine and CADs [16]. The localization and tissue concentrations of active metabolites differ among the echinacea species studied so far and undergo temporal changes, including seasonal variations and fluctuations related to plant phenology [14].

## 2. Bioactive Metabolites of Echinacea

Secondary metabolites are compounds that plants produce as an adaptation response to various biotic and abiotic stresses [4]. Echinacea species contain a variety of compounds, including alkamides, CAD esters (such as chicoric acid), polysaccharides, and polyacetylenes (Table 2). Several biological activities have been identified for the chemical components found in echinacea species [14]. For example, the polysaccharides enhance macrophage activity and various processes related to cytokine synthesis [16]. Specific groups of phenolic compounds and alkamides present in echinacea tissues have demonstrated antiviral and antifungal activities [17].

The combined effects of various bioactive substances may contribute to the biological activities of echinacea [14,16]. However, it is worth noting that both CADs and alkamides play a significant role in most biological activities and pharmacological effects attributed to echinacea products [18], such as antitumor, antioxidant, antimicrobial, antifungal, antiviral, and immunomodulatory effects (Table 3). To ensure consistency before conducting clinical trials, it is essential to standardize echinacea products based on their content of the specific marker compounds [14]. This standardization process helps maintain product quality and reliability.

### 2.1. Alkylamides

Alkylamides are natural compounds formed when various amines combine with straight-chain aliphatic acids (typically unsaturated) through amide bonds [19]. These compounds are diverse, encompassing more than 300 identified compounds from eight plant families, including various combinations of 200 acids and 23 amines [17]. Alkylamides fall into two groups: purely olefinic patterns and mixed olefinic–acetylenic types, with some saturated derivatives [14]. Alkylamides are primarily found in the roots and aerial parts of echinacea species (Table 4), and some researchers have proposed these compounds as the key bioactive components of echinacea [20]. Alkylamides were detected in the 80% *v*/*v* ethanol extract of *E. purpurea* roots using high-performance liquid chromatography (HPLC) [21].

A diverse range of biological activities, such as immunomodulation, antibacterial, antiviral, larvicidal, insecticidal, diuretic, analgesic, and antioxidant effects, have been attributed to alkylamides [22,23]. The alkylamides also exhibit the capacity to inhibit processes like prostaglandin biosynthesis, RNA synthesis, and arachidonic acid metabolism; in addition, they can enhance the effectiveness of several antibiotics [15,24] and are considered to possess both immunostimulatory and anti-inflammatory properties [25].

Alkylamides interact with the cannabinoid receptor type 2, potentially explaining their immunomodulatory effects [26]. Research indicates that N-alkamides from *E. purpurea* roots stimulate interleukin-10 (IL-10) and inhibit tumor necrosis factor-alpha (TNF-α) in vitro [14]. Administering the alkylamide fraction from *E. purpurea* roots to healthy rats exposed to bacterial lipopolysaccharide (LPS) increased nitric oxide (NO) and TNF-α release from alveolar macrophages and boosted phagocytic activity [8]. Importantly, alkylamides in *E. purpurea* lozenges were swiftly absorbed through the buccal and esophageal membranes in six healthy volunteers [14].

**Table 2 plants-13-01235-t002:** Chemical constituents and their concentrations in echinacea roots.

Class	Concentration (%)	Chemical Compounds	References
Alkylamides	0.01–0.70 (*w*/*w*)	Isobutyl amides of straight-chain fatty acids with olefinic and/or acetylenic bonds, e.g., isomeric dodeca-2E,4E,8Z,10E/Z-tetraenoic isobutylamide, undeca-2Z,4E-diene-8,10-diynoic acid isobutylamide	[24]
Caffeic acid	2.0–2.8 (*w*/*w*)	Chicoric acid (2,3-*o*-di-caffeoyl tartaric acid) derivatives of 2-*o*-caffeoyltartaric acid, echinacoside, verbascoside, caffeoylechinacoside, and chlorogenic acid	[25]
Polysaccharides	1.5–2.5	Arabinogalactans and glycoproteins contain a sugar component comprising arabinose, galactose, and galactosamine	[19]
Volatile oils	0.1	Caryophyllene, caryophyllene oxide, humulene, limonene, camphene, aldehydes, and dimethyl sulfide. As per WHO monograph, pentadeca-(1,8-Z)-diene, 1-pentacene, ketoalkynes, and ketoalkenes are also present	[27]
Others	-	Small amounts of polyacetylene compounds including trideca-1-en-3,5,7,9,11-pentane, trideca-1,11-dien-3,5,7,9-tetrayne, trideca-8,10,12-triene-2,4,6-triene. Effective alkaloids: tussilagine and isotussilagine.	[28]

**Table 3 plants-13-01235-t003:** Biological and pharmacological effects of the bioactive compounds of echinacea roots.

Bioactive Compounds	Biological and Pharmacological Effects	References
Alkylamides	Immunomodulators, anti-inflammatory, macrophage modulation, decrease in NO and tumor necrosis factor-α, antiviral immunity mediators, and type 2 cannabinoid receptor	[29]
Polysaccharides	Antitumor, antioxidant, antimicrobial, antifungal, antiviral, immunomodulators, hypoglycemic, hepatoprotective, gastrointestinal protector, and anti-diabetic	[30]
Glycoproteins	Immunomodulatory	[31]
Flavonoids	Antioxidants, anti-inflammatory, anti-ulcer, anti-allergic, and antiviral activity	[32]
CADs	Anti-inflammatory, antioxidant, anti-osteoporotic, antimicrobial, antitumor, and neuroprotective activity	[33]

**Table 4 plants-13-01235-t004:** Most abundant alkylamides in roots of echinacea species.

Alkylamide Compounds	Molecular Weight (g/mol)	Echinacea Species	Reference
Undeca-2E,4Z-diene-8,10-diynoic acid isobutylamide	229.32	*E. purpurea*, *E. angustifolia*	[13]
Undeca-2Z,4E-diene-8,10-diynoic acid isobutylamide	229.32	*E. purpurea*	[3,16]
Undeca-2E-ene-8,10-diynoic acid isobutylamide	231.34	*E. purpurea*	[24]
Undeca-2E,4Z-diene-8,10-diynoic acid 2-methylbutylamide	243.35	*E. purpurea*	[6]
Undeca-2Z,4E-diene-8,10-diynoic acid 2-methylbutylamide	243.35	*E. purpurea*	[1]
Dodeca-2Z,4E-diene-8,10-diynoic acid isobutylamide	243.35	*E. purpurea*	[14]
Dodeca-2E,4Z-diene-8,10-diynoic acid isobutylamide	243.35	*E. purpurea*	[10]
Dodeca-2E,4E,10E-triene-8-ynoic acid isobutylamide	245.37	*E. purpurea*	[2]
Dodeca-2E-ene-8,10-diynoic acid isobutylamide	245.37	*E. angustifolia*	[34]
Dodeca-2E,4E,8Z,10E-tetraenoic acid isobutylamide	247.38	*E. angustifolia*	[4]
Dodeca-2E,4E, 8Z,10Z-tetraenoic acid isobutylamide	247.38	*E. purpurea*	[9]
Dodeca-2E,4E, 8E,10Z-tetraenoic acid isobutylamide	247.38	*E. purpurea*	[21]
Dodeca-2E,4E,8Z-trienoic acid isobutylamide	249.40	*E. purpurea*	[10]
Dodeca-2E,4E-dienoic acid isobutylamide	251 41	*E. purpurea*	[8]
Trideca-2E,7Z-diene-8,10-diynoic acid isobutylamide	257.38	*E. purpurea,*	[13]
Dodeca-2E,4Z-diene-8,10-diynoic acid 2-methylbutylamide	257.38	*E. purpurea*	[24]
Dodeca-2,4,8,10-tetraenoic acid 2-methylbutylamide	261.41	*E. purpurea*	[35]

### 2.2. Caffeic Acid Derivatives

A variety of CADs are known to exist in echinacea, including caftaric acid, chlorogenic acid, caffeic acid, cynarin, echinacoside, and chicoric acid [27] (Figure 1). Table 5, Table 6, Table 7 and Table 8 provide a summary of the contents of chicoric acid, echinacoside, chlorogenic acid, and cynarine in different organs of three echinacea species, and how these measurements can be affected by factors such as extraction methods and analytical techniques. Recent efforts have focused on producing CAD compounds from adventitious root cultures [17]. Among the CADs, chicoric acid is considered a major constituent in *E. purpurea*, found abundantly in the root and petiole tissues [25], with concentrations ranging from 1.2% to 3.1% of dry weight in roots and from 0.6% to 2.1% of dry weight in flowers [7]. Caffeic acid is commonly found in *E. purpurea* as chlorogenic acid, an ester of quinic acid [3]. Like other polyphenols, CADs can offer various health benefits due to their antioxidant properties, which may include reducing the risk of conditions such as diabetes, cancer, and neurological diseases [19].

Caffeic acid has diverse biological benefits, including antioxidant, anti-inflammatory, anticancer, and neuroprotective effects [27], and shows therapeutic potential by regulating transcription and growth factors, as evidenced in studies with human cell cultures and animal models [25]. The use of naturally occurring bioactive compounds like CADs is gaining popularity in modern societies [14], highlighting the importance of understanding their characteristics and roles.

Among CADs, chicoric acid and echinacoside are considered the most significant compounds due to their major pharmacological properties [24]. For instance, Soltanbeigi and Maral (2022) found that chicoric acid protects collagen, inhibits HIV-1, boosts phagocyte activity, and scavenges free radicals. Echinacoside, another key CAD, is well known for its immunostimulatory and potent antioxidant activities [19]. For chlorogenic acid, another important dietary supplement, suggests it prevents type 2 diabetes by inhibiting glucose absorption and has anticarcinogenic effects by countering damage from carcinogenic N-nitroso compounds. Cynarin possesses cholagogue and choleretic properties, whereas caftaric acid is a potent antioxidant [19]. Among various CADs, echinacoside and chicoric acid are the most commercially significant dietary supplements.

### 2.3. Flavonoids

Echinacea species (e.g., *E. purpurea*, *E. angustifolia*, and *E. pallida*) contain a variety of flavonoids, a group of plant metabolites known for their potential health benefits through modulating cell signaling pathways and having antioxidant effects [28]. Quercetin and kaempferol are the two key flavonoid compounds found in most echinacea species. Quercetin is associated with several health benefits, including potential anticancer, anti-inflammatory, and antiviral properties [29]. Kaempferol, found in echinacea species including *E. angustifolia*, has potential antioxidant and anti-inflammatory properties [17].

**Table 5 plants-13-01235-t005:** Chicoric acid content in three echinacea species as influenced by plant organ and extraction technique.

Plant Organ	Extraction Technique	Analysis	*E. purpurea*	*E. pallida*	*E. angustifolia*	Reference
Root	Ethanol 70% * (shaker)	HPLC	22.7 mg g^−1^ DW		0.9 mg g^−1^ DW	[36]
Root	Ethanol 70%	HPLC	9.4 mg g^−1^ DW	0.5 mg g^−1^ DW	0.1 mg g^−1^ DW	[37]
Root	Methanol 70%	HPLC-ESIMS	11.0 mg g^−1^ DW			[38]
Root	Methanol 70%	HPLC-ESIMS	20.8 mg g^−1^ DW			[39]
Root	Methanol 80%	HPLC	19.3 mg g^−1^ DW	0.83 mg g^−1^ DW	<LOQ **	[40]
Root	Ethanol 55%	HPLC	4.77 mg g^−1^ DW	0.032 mg mL^−1^	0.046 mg mL^−1^	[41]
Root	Ethanol 80%	HPLC	13.6 mg g^−1^ DW			[42]
Shoot	Ethanol 70%	HPLC	6.00 mg g^−1^ DW			[43]
Root	Methanol 80%	HPLC	19.0 mg g^−1^ DW	0.41 mg g^−1^ DW	0.27 mg g^−1^ DW	[44]
Root	Ethanol 50%	HPLC	0.71 mg g^−1^ DW			[45]
Hairy root	Methanol 70%	HPLC-ESIMS	19.2 mg g^−1^ DW			[46]
Root	Methanol 70%	HP LC	25 mg g^−1^ DW			[47]
Root	Methanol 70% (sonication)	HPLC	7.63 mg g^−1^ DW			[48]
Root	Methanol 70%	HPLC	7.7 mg g^−1^ DW			[49]
Root	Methanol 70% (sonication)	HPLC	8.17 mg g^−1^ DW			[50]
Shoot	Methanol 70% (sonication)	HPLC	8.57 mg g^−1^ DW			[51]
Root	Methanol 70% (sonication)	HPLC	41.3 mg g^−1^ DW			[52]
Root	Ethanol 70%	HPLC	11.3 mg g^−1^ DW			[53]
Root	Methanol and water	HPLC	13.9 mg g^−1^ DW			[54]
Steam	Methanol and water	HPLC	16.7 mg g^−1^ DW			[55]
Leaves	Methanol 70% (sonication)	HPLC	20.2 mg g^−1^ DW			[56]

* All % values are *v*/*v*; ** LOQ, limit of quantification.

**Table 6 plants-13-01235-t006:** Echinacoside content in three echinacea species as influenced by plant organ and extraction technique.

Plant Organ	Extraction Technique	Method	*E. purpurea*	*E. pallida*	*E. angustifolia*	Reference
Root	Ethanol 70% * (shaker)	HPLC	<0.1 mg g^−1^ DW	3.4 mg g^−1^ DW	10.4 mg g^−1^ DW	[57]
Root	Ethanol 70%	HPLC		3.7 mg g^−1^ DW	3.6 mg g^−1^ DW	[58]
Root	Methanol 80%	LC-MS	<LOQ **	16.2 mg g^−1^ DW	9.10 mg g^−1^ DW	[59]
Root	Ethanol 60% (ultrasonic)	HPLC			0.245 mg mL^−1^	[60]
Root	Ethanol 55%	HPLC	0	0.62 mg mL^−1^	1.86 mg mL^−1^	[61]
Root	Methanol 80% (stirring)	HPLC	<LOQ	12.7 mg g^−1^ DW	10.6 mg g^−1^ DW	[62]
Shoot	Ethanol 70%	HPLC	0			[63]
Flower	Ethanol 70%	HPLC	0			[64]
Herb	Ethanol 70%	HPLC	0			[65]
Root	Methanol 70% (ultrasonic)	HPLC	1.1 mg g^−1^ DW			[66]
Shoot	Ethanol 70% (sonication)	HPLC	0.5 mg g^−1^ DW			[67]

* All % values are *v*/*v*; ** LOQ, limit of quantification.

**Table 7 plants-13-01235-t007:** Chlorogenic acid content in three echinacea species as influenced by plant organ and extraction technique.

Plant Organ	Extraction Technique	Method	*E. purpurea*	*E. pallida*	*E. angustifolia*	Reference
Root	Ethanol 70% *	HPLC	<0.1 mg g^−1^ DW	0.3 mg g^−1^ DW	1.5 mg g^−1^ DW	[68]
Root	Methanol 80%	HPLC	<LOQ **	<LOQ	<LOQ	[69]
Root	Methanol 60%	HPLC	0.29 mg g^−1^ DW			[70]
Root	Methanol 80%	HPLC	<LOQ	<LOQ	0.77 mg g^−1^ DW	[71]
Shoot	Ethanol 70%	HPLC	0.045 mg mL^−1^			[72]
Root	Ethanol 55%	HPLC	0.055 mg mL^−1^	0.003 mg mL^−1^	0.282 mg mL^−1^	[73]
Flower	Ethanol 70%	HPLC	0.208 mg mL^−1^			[74]
Hairy root	Methanol 70%	HPLC	0.93 mg g^−1^ DW			[75]
Root	Methanol and water	HPLC	0.011 mg g^−1^ DW			[76]
Shoot	Methanol and water	HPLC	0.152 mg g^−1^ DW			[77]
Shoot	Ethanol 70%	HPLC	0.3 mg g^−1^ DW			[78]

* All % values are *v*/*v*; ** LOQ, limit of quantification.

**Table 8 plants-13-01235-t008:** Cynarin content in three echinacea species as influenced by plant organ and extraction technique.

Plant Organ	Extraction Technique	Method	*E. purpurea*	*E. pallida*	*E. angustifolia*	Reference
Root	Ethanol 70% *	HPLC	<0.1 mg g^−1^ DW	<0.1 mg g^−1^ DW	1.2 mg g^−1^ DW	[79]
Root	Methanol 80%	HPLC	<LOQ **	<LOQ	1.39 mg g^−1^ DW	[80]
Root	Methanol 60%	HPLC	0.8 mg g^−1^ DW			[81]
Root	Ethanol 60%	LC-MS			0.09 mg mL^−1^	[82]
Flower	Ethanol 70%	HPLC	0			[83]
Herb	Ethanol 70%	HPLC	0			[84]
Shoot	Ethanol 70%	HPLC	0.005 mg mL^−1^			[85]
Root	Ethanol 55%	HPLC	0	0	0.238 mg mL^−1^	[86]
Root	Methanol 80%	HPLC	<LOQ	<LOQ	3.44 mg g^−1^ DW	[87]
Root	Ethanol 70%	HPLC	0.13 mg g^−1^ DW			[88]
Shoot	Ethanol 70%	HPLC	0.2 mg g^−1^ DW			[89]

* All % values are *v*/*v*; ** LOQ, limit of quantification.

### 2.4. Polysaccharides

Several modification methods are used to explore polysaccharide derivatives in recent echinacea research, including sulfation, acetylation, phosphorylation, carboxymethylation, amination, benzoylation, C-glycosylation, hydroxypropylation, and selenization [30,31]. It is reported that polysaccharide hydroxyl groups can undergo etherification and esterification, whereas those with uronic acid can engage in nucleophilic and electrophilic reactions, such as esterification and amide formation [32].

More polysaccharide content was found in echinacea species with green stems, such as *E. purpurea*, compared to those with red stems, with flowers having higher content than leaves [26]. The main polysaccharides present in *E. purpurea* are outlined in Table 9. A 0.1 mg mL^−1^ *E. purpurea* extract demonstrated a 30% greater antioxidant capacity than ascorbic acid at the same concentration [31].

## 3. Pharmacological Advantages of Echinacea Phytochemicals

In the realm of medicinal application, three distinct echinacea species are harnessed, each ascribed with unique therapeutic attributes. Despite this perceived divergence, there is a notable dearth of comparative research assessing the efficacy of these species [33]. Even though the composition of these herb species bears similarities, nuanced fluctuations in the levels of active components are discernible, owing to factors such as geographic location, developmental stage, harvest timing, and growth conditions [35]. The utilization of echinacea for dietary supplements extends to the roots, leaves, or the entire plant. Significantly, the root composition sharply contrasts with that of the aboveground plant parts, with elevated concentrations of volatile oils and pyrrolizidine alkaloids (including tussilagine and isotussilagine) in roots, whereas the aboveground plant parts boast active constituents such as caffeic and ferulic acid derivatives (e.g., chicoric acid and echinacoside) and specific polysaccharides (e.g., acidic arabinogalactan, rhamnoarabinogalactan, and 4-*o*-methylglucuronylarabinoxylans) [46]. Numerous other active components in echinacea have been cataloged, yet the relative potency, potential synergistic effects, and bioavailability of these compounds when ingested remain enigmatic [36].

### 3.1. Neuroprotective Effect

Echinacoside has exhibited significant neuroprotective efficacy (Figure 2, Table 10). The concept of neuroprotection involves a therapeutic strategy dedicated to shielding neurons from demise, hampering the progression of diseases, and prolonging the shift from preclinical to clinical stages [37]. This entails the capacity to impede or defer neuronal death by intervening in neurodegenerative processes, whether occurring prematurely or in the aging process [36]. Neuroprotective interventions are approved for the central nervous system disorders like Parkinson’s and Alzheimer’s disease, with growing interest in the echinacoside potential impact [38]. The neuroprotective effects of echinacoside are primarily associated with its action in pathways involving apoptosis and neuroinflammation [39]. This underscores its promising role in addressing neurological challenges and warrants further investigation into its mechanisms and therapeutic potential in neurodegenerative disorders.

The pathogenesis of Parkinson’s disease (PD) is closely linked to the inflammatory response orchestrated by activated microglia. In a 1-methyl-4-phenyl-1,2,3,6-tetrahydropyridine (MPTP) mouse model, the administration of echinacoside led to a notable decrease in the expression of the microglial marker ionized calcium-binding adapter molecule-1 (Iba-1) in the midbrain. Furthermore, echinacoside treatment demonstrated an effective dampening of glial cell activation, contributing to the amelioration of brain inflammation. These findings shed light on the promising anti-inflammatory properties of echinacoside and suggest its potential as a therapeutic agent in attenuating neuroinflammation associated with PD, thereby presenting a prospective avenue for intervention in neurodegenerative conditions.

These mechanisms effectively suppressed inflammation in dopaminergic neurons within the midbrain, indicating a notable neuroprotective effect [41]. Additionally, echinacoside displayed substantial protective effects against TNF-α-induced apoptosis in SH-SY5Y neuronal cells. These effects were attributed to its antioxidant properties that mitigate mitochondrial dysfunction, reduce intracellular generation of reactive oxygen species, inhibit caspase-3 activity, and maintain a high mitochondrial membrane potential [42]. With strong anti-apoptotic activity, echinacoside emerges as a promising therapeutic agent for diverse neurodegenerative and neurological conditions characterized by neuronal apoptosis. Earlier studies have shown that echinacoside possesses the capability to avert the decline of dopamine and its metabolites in the extracellular fluid of the striatum in rats experiencing acute injury induced by 6-hydroxydopamine. This effect is attributed to echinacoside’s capacity to reduce the generation of free radicals and suppress oxidative stress [35]. Furthermore, echinacoside has been observed to boost the insertion of α-amino-3-hydroxy-5-methyl-4-isoxazolepropionic acid receptor (AMPAR) into the membrane and elevate the expression of brain-derived neurotrophic factor (BDNF), which may be associated with neuroprotection [40]. Thus, the multifaceted and robust therapeutic effects of echinacoside position it as a highly promising and versatile candidate for pharmaceutical intervention in Parkinson’s disease and related neurological disorders, offering potential avenues for clinical applications.

### 3.2. Anticancer Activity

Echinacoside emerges as a compelling candidate in the ongoing battle against cancer, a pervasive global health challenge that continues to claim a significant toll on mortality rates. The wealth of evidence from various in vitro and in vivo studies, as illustrated in Figure 3 and detailed in Table 11, consistently underscores the anticancer properties of echinacoside. Its efficacy extends across diverse cancer types, including colorectal [43], breast [41], and liver [44]. Mechanistically, echinacoside exerts its anticancer effects by hindering excessive cell proliferation, suppressing invasive and migratory processes, inducing cell cycle arrest, and fostering apoptotic pathways [45]. These multifaceted anticancer properties position echinacoside as a promising therapeutic agent with the potential to address a spectrum of malignancies, highlighting its significance in the pursuit of novel cancer treatment strategies.

Regarding colorectal cancer, echinacoside shows its effectiveness by arresting SW480 cells at the G1 phase, a process mediated through the activation of the mitochondria-associated intrinsic apoptosis pathway, ultimately culminating in caspase-dependent apoptosis [46]. The cell cycle arrest is linked to an increase in CDKN1B (p21) expression, a key G1/S-CDK blocker and DNA synthesis inhibitor. Additionally, induction of apoptosis involves downregulating the anti-apoptotic protein Bcl-2, upregulating the pro-apoptotic protein Bax, and decreasing mitochondrial membrane potential [41]. Despite these promising mechanisms, the limited bioavailability and penetration of echinacoside into enterocytes hinder the direct targeting of disseminated tumor cells in circulation or distant organs [46]. However, adjusting echinacoside dosage and timing can enhance growth of *Faecalibacterium prausnitzii*, a common SCFA producer. Live *F. prausnitzii* strains have shown promise in suppressing metastasis in vivo, suggesting the echinacoside potential as an antimetastatic agent targeting gut microbiota [47]. The growing in vitro evidence supporting the echinacoside antitumor and antimetastatic effects highlights its promise as a therapeutic agent for colorectal cancer.

Echinacoside may have therapeutic potential in the treatment of breast cancer. It diminishes the expression of key proteins, including photo-LRP6, total LRP6, photo-Dvl2, active β-catenin, and total β-catenin, effectively suppressing the Wnt/β-catenin signaling pathway. This modulation downregulates Wnt target genes, implicating the inhibition of the Wnt/β-catenin pathway as a plausible mechanism for the echinacoside preventive impact on breast cancer, supported by both in vitro and in vivo studies [48]. Additionally, echinacoside inhibits breast cancer cells by downregulating miR-4306 and miR-4508 expression, suppressing cell proliferation, invasion, and migration, while promoting apoptosis [49]. The multifaceted mechanisms of action of echinacoside, coupled with its demonstrated low toxicity levels, underscore its considerable significance in cancer treatment, particularly as a versatile and promising therapeutic agent for breast cancer.

Liver cancer is characterized by the escalating incidence rates of hepatocellular carcinoma (HCC), which constitutes a substantial majority of cases [40]. The activation of AKT (p-AKT) is a notable prognostic marker in HCC, showing a correlation with the disease invasiveness and predicting unfavorable outcomes [35]. Echinacoside shows strong effectiveness against liver cancer by reducing the levels of the triggering receptors expressed on myeloid cells 2 (TREM2) and influencing the phosphatidylinositol-3-kinase/protein kinase B (PI3K/AKT) signaling pathway [47]. Additionally, it plays a key role in liver cancer development by affecting the transforming growth factor-β1 (TGF-β1)/Smad signaling pathway. Intriguingly, it downregulates TGF-β1 and Smad3 while upregulating Smad7, thereby effectively inhibiting the activation of the TGF-β1/Smad signaling pathway [49]. These comprehensive insights underscore the multifaceted mechanisms through which echinacoside exerts its antitumor effects in liver cancer, suggesting its potential as a valuable therapeutic intervention in this challenging malignancy.

### 3.3. Liver-Protective Efficacy

Liver, the body’s largest organ, plays a crucial role in detoxification, immune support, and metabolism regulation [50]. Liver diseases, including viral hepatitis, fatty liver, and liver cancer, impose significant global health burdens [50]. According to the GLOBOCAN 2020 database, 905 million individuals worldwide are affected by chronic liver diseases, resulting in approximately 830 million deaths attributed to liver-related issues. Despite advancements in pharmaceuticals and technologies, the prognosis for end-stage liver diseases remains grim. Herbal medicine offers a promising approach for both prevention and treatment, with favorable outcomes and minimal side effects [33], underscoring its potential as a valuable adjunct in addressing liver-related ailments globally. Echinacoside, a naturally occurring water-soluble phenylethanoid glycoside, emerges as a promising agent in preventing and treating a diverse array of liver disorders, showing notable hepatoprotective effects by thwarting various forms of liver injury [51,52]. Its impact extends to drug metabolism by inhibiting key cytochrome P450 enzymes (CYPs), including CYP1A2, CYP2E1, CYP2C19, and CYP3A4, which are crucial in the oxidative metabolism of clinical drugs [53]. In the context of drug-/chemical-induced liver injury (a significant clinical concern globally), echinacoside proves effective in enhancing antioxidant enzyme activities, mitigating oxidative stress, and suppressing pro-inflammatory cytokines like IL-1β, IL-6, and TNF-α [36,52]. Furthermore, its protective effects extend to ethanol-induced liver injury, involving the alleviation of oxidative stress, reduction in cell apoptosis through Nrf2 upregulation, and modulation of the SREBP-1c/FASN pathway via PPAR-α, highlighting its potential as a multifaceted therapeutic agent supporting liver health [35].

Globally, approximately 2 billion people are infected with the HBV virus, highlighting the widespread prevalence of hepatitis B. In China, about 10% of the population are carriers [54]. Echinacoside showcases significant inhibitory effects on HBsAg and HBeAg expression in HBV transgenic mice, coupled with a notable reduction in serum HBV DNA levels, suggesting its potential as a therapeutic agent in managing hepatitis B progression [55]. Failure to promptly address the causative factors may lead to the advancement of the diseases to severe stages, including fibrosis, cirrhosis, and eventual liver cancer [50]. Particularly noteworthy is the echinacoside anti-fibrotic efficacy associated with disrupting signaling pathways in TGF-β1/Smad and inhibiting activation of hepatic stellate cells, positioning echinacoside as a promising herbal medicine for liver fibrosis treatment [56]. Additionally, its antitumor activity through the PI3K/AKT and TGF-β1/Smad signaling pathways further reinforces its potential as a versatile therapeutic agent for various liver diseases [36,49] (Table 12).

## 4. Recent Advances in Biotechnology for Echinacea Production

### 4.1. Genetic Engineering and Phylogenetic Analysis of Echinacea

The efforts to develop “improved” varieties are ongoing. Echinacea breeding studies and patents primarily focus on ornamental aspects and reducing seed dormancy. Traditional selective breeding taps into genetic and phenotypic diversity in cultivated and wild echinacea plants, gaining acceptance from the public and organic farming industry. Molecular genetic techniques offer a precise approach to modifying developmental and biosynthetic processes by directly altering a plant genome [57]. Despite public GMO concerns, potentially “organically acceptable” biotechnological approaches for echinacea modification, such as transformation with Agrobacterium and polyploid induction, have been developed [58]. Isozyme data have played a pivotal role in phylogenetic analyses involving four echinacea taxa, namely *E. tennesseensis*, *E. angustifolia*, *E. pallida*, and *E. purpurea*. *E. tennesseensis*, confined to limestone glades in Middle Tennessee, is considered to be closely affiliated with the more widespread prairie species *E. angustifolia* [59]. Some taxonomists amalgamate the first three taxa into a single species (*E. pallida*), whereas a consensus persists that *E. purpurea* exhibits distinct morphological characteristics [60]. Consistent findings from various phylogenetic reconstruction methods underscore the separation of populations of *E. tennesseensis*, *E. angustifolia*, and *E. purpurea* into distinct lineages [61]. However, the phylogenetic relationship of *E. pallida* populations to the other taxa remains unresolved in the existing data [62]. Notably, the reported mean genetic identity between *E. tennesseensis* and *E. angustifolia* is somewhat lower than the mean genetic identity for either taxon compared with *E. pallida* [63]. Despite the perceived closer relationship between the first two species, the discussion of various evolutionary hypotheses continues. Nevertheless, definitive conclusions hinge on a comprehensive phylogenetic analysis of the entire echinacea genus employing higher-resolution data [64].

To gain a more thorough grasp of the population genetics within both *E. pallida* and *E. purpurea*, an extensive sampling strategy covering the entirety of their ranges is recommended, considering the evident heterogeneity present [65]. The question of how well the disjunct Tennessee populations represent *E. pallida* as a whole remains unresolved and necessitates further exploration [66]. The substantial divergence observed in *E. purpurea* populations (marked by the greatest dissimilarity) is expected, given their smaller size compared to populations of other taxa, making them more susceptible to the influences of random genetic drift [67]. It is noteworthy that *E. tennesseensis* has been proposed as having a close relationship with *E. angustifolia* var. *angustifolia* [68]. Previous research [67,68] has documented morphological and anatomical differences among all echinacea taxa, with some subtle differences. Importantly, there appear to be no substantial internal barriers impeding gene flow, even among the most morphologically distinct echinacea species [65]. In a comprehensive crossing program involving all species except *E. tennesseensis*, the previous report [68] also found evidence of the interfertility of all taxa. The divergence between *E. tennesseensis* and *E. angustifolia* is presumed to have originated from a common ancestor through geographic isolation, a process likely occurring during a period when prairies extended further east than today, and glade vegetation was more prevalent in Tennessee [57]. This historical context provides valuable insights into the evolutionary dynamics shaping these echinacea species (Figure 4).

It is reported that coding and non-coding regions of chloroplast genomes delineate the phylogeny of echinacea species [65]. The nine species formed two well-supported clades. The first clade included *E. tennesseensis*, *E. speciosa*, *E. purpurea*, and *E. laevigata*, with *E. tennesseensis* closely linked to *E. speciosa*, forming a sister group to *E. purpurea*. *E. laevigata* showed a close relationship with these three species. The second clade comprised five species, with *E. angustifolia* closely associated with the others. Within this clade, *E. atrorubens* was a sister to *E. paradox*, and *E. pallida* was a sister to *E. sanguinea* (Figure 5A,B).

Table 13 outlines the base differences among the nine echinacea species, with the number of differences ranging from 181 to 910. The top 25 variable non-coding regions and their overlaps with previous studies for low-level phylogenetic inferences are shown in Table 14. The alignment suggests that the intergenic region between trnH and psbA could serve as DNA barcoding for most echinacea species when combined with ITS. However, validation with more individuals is advised due to the reliance on limited SNPs. The trnH-psbA PCR product varies in size, with SNPs between species ranging from 0 to 16. Universal primers for trnH-psbA should successfully amplify all species. Additionally, the ITS marker could aid in the differentiation of species pairs not distinguishable with trnH-psbA alone. However, diagnostic SNP numbers remain low, and bootstrap values for constructed trees are minimal.

### 4.2. In Vitro Technologies for Mass Propagation of Echinacea

The utilization of in vitro culture and plant regeneration offers distinct advantages over traditional vegetative propagation, primarily attributed to the significantly accelerated rate of plant multiplication [60]. Furthermore, this method can be particularly effective in propagating species that may exhibit lower responsiveness to conventional cloning techniques [71]. Notably, echinacea species have demonstrated successful regeneration from diverse tissues, ranging from in vitro seedlings to mature, field-grown plants.

### 4.3. In Vitro Seed Germination

Due to its predominant organic cultivation [72], echinacea, including its seeds, is susceptible to substantial microbial contamination [73,74]. Seeds play a crucial role as explants in establishing in vitro cultures of echinacea [71]. Various seed sterilization methods have been employed, such as surface sterilization with ethanol and sodium hypochlorite, supplemented with the detergent Tween 20 [75]. However, relying solely on surface sterilization may not eliminate microbial contamination [72]. To address this, the use of a broad-spectrum antimicrobial agent, such as Plant Preservation Mixture (PPM) from Phytotechnology Laboratories in Lexena, Kansas, United States, has been employed to control systemic fungal contamination of echinacea seeds, ensuring the production of sterile seedlings [72,75].

Harbage (2001) proposed removing seed coat layers to prevent contamination. For *E. purpurea* seeds, a surface sterilization process involving immersion in 10% *w*/*w* PPM, 70% *v/v* ethanol for 30 s, and 5.4% *v/v* sodium hypochlorite with traces of Tween 20 for 18 min led to contamination-free germination [60]. Basal media components alone suffice for in vitro echinacea seed germination, with seed explants developing shoots when exposed to cytokinins [76]. Recent studies suggest that endophytic, antibiotic-resistant bacteria can coexist with echinacea cultures without harming plant growth [77].

### 4.4. Explants

The selection of explants varies among species and is a crucial factor in determining propagation efficiency. Numerous regeneration methods have been reported for commercially relevant echinacea species, with almost all protocols involving the use of embryonic or in vitro-grown seedling explants [78,79,80,81]. Various plant tissues, including anther, mesophyll protoplasts, petiole, stem, seed, flower stalks, leaf sections, hypocotyls, cotyledons, and roots, were initially used as explants to induce formation of callus that subsequently underwent differentiation into shoots and roots [82]. The preference for juvenile tissues in these choices of explant material is based on their generally high organogenic competence [82]. In vitro, echinacea regeneration from leaf tissue is less invasive than that from embryonic sources [82]. Due to genetic uncertainties with seed or seedling tissues, leaves are preferred for regeneration. Importantly, the same explant can exhibit varied responses under different cultural conditions [60,72].

### 4.5. Shoot Organogenesis

Biochemical processes play a pivotal role in plant shoot morphogenesis [83]. Generally, the type of explant, its orientation in the culture medium, and the presence of plant growth regulators play key roles in regulating the differentiation process [84]. Koroch et al. (2002) [59] induced callus and indirect shoot organogenesis from *E. purpurea* leaf explants using different NAA/BAP combinations. BAP alone at lower concentrations stimulated shoot formation and increased callus production but showed low shoot initiation when combined with increased NAA concentrations.

BAP combined with NAA induced shoot growth in *E. pallida* [82], whereas NAA with TDZ or BAP was effective for shoot development in *E. tennesseensis* [85]. Lower BAP concentrations (0.45–4.5 μmol/L) promoted shoot growth from seed explants [76]. Coker and Camper (2000) favored NAA and kinetin over 2,4-D and kinetin. Petiole explants of *E. purpurea* showed shoot organogenesis when treated with BAP or TDZ combined with IAA [72].

### 4.6. Somatic Embryogenesis

Somatic embryogenesis, seen notably in *E. purpurea* petiole explants treated with BAP, TDZ, or a combination of TDZ and IAA, results in a high yield of plantlets [72]. Microscopic analysis revealed a well-defined protoderm without vascular connections to the maternal vasculature. Even though well-defined embryos were observed in all cultures, *E. angustifolia* and *E. pallida* exhibited greater embryogenic potential than *E. paradoxa* and *E. purpurea*. Additionally, leaf disk culture induced both de novo shoots and somatic embryos, with considerable variation in regeneration modes depending on the seedling-derived line origin.

The regenerative response in echinacea is significantly influenced by the selection of explant sources, culture methods, genetic backgrounds, and tissue conditions [87]. Hypocotyls often exhibit heightened responsiveness compared to other tissue types in multiple echinacea species [71,87], with genotype-related differences in embryogenic capacity noted across species and cultivars [71]. Despite extensive research, the regulatory factors governing plant cell morphogenic competence remain elusive. It is increasingly evident that different explants and even cells within the same explant possess distinct states of morphogenic competence, requiring diverse cues to initiate specific regenerative pathways [78]. Consequently, the varying efficiencies of explants in response to auxin and cytokinin combinations reflect unique states of competence, necessitating different inductive signals for specific regenerative responses [71,72].

### 4.7. Regeneration of Protoplasts

The application of cell manipulation techniques, including somaclonal variation and somatic hybridization through protoplasts, stands as a promising approach for the development of novel and enhanced echinacea cultivars. Previous research has successfully isolated protoplasts from various echinacea tissues using enzymatic digestion methods [88]. These protoplasts, particularly from *E. purpurea* mesophyll tissues, have been utilized in a plant regeneration system employing an alginate-embedding culture technique, leading to cell colony formation, callus proliferation, and shoot organogenesis in response to common auxin and cytokinin combinations [80]. Protoplast fusion, especially across closely and distantly related species, introduces the possibility of bulk DNA transfer, a departure from conventional genetic transformation methods focusing on one or two genes [82]. Despite an anticipated revolution in variety development, the practical use of protoplast fusion has faced challenges in regenerating plants from fused protoplasts [78]. Nonetheless, somatic hybridization between closely related echinacea species has yielded practical benefits, potentially creating unique germplasm with synergistic effects of various medicinal compounds [72]. Further optimization of protoplast regeneration and fusion processes, both within and across different genotypes and species, promises significant advancements in fundamental research and the development of novel commercial products.

## 5. Stress-Induced Enhancement of Bioactive Compounds

Secondary metabolites play a critical role in plant response to various types of biotic stresses (e.g., pathogens) and abiotic stresses including temperature extremes, drought, salinity, and exposure to UV light [89]. The plant defense mechanisms are modulated by transcriptional factors (TFs) through detecting stress signals and regulating the expression of downstream defense genes [27]. Furthermore, the survival, resilience, and productivity of plants are contingent on the increased synthesis of secondary metabolites, driven by a process known as elicitation [29].

Hydroponic systems are essential for efficiently mass-producing echinacea plants. The upcoming sections explained how salinity and extreme temperatures can boost the production of bioactive compounds in echinacea species.

### Salt Stresses

Salinity, a major abiotic stressor, can cause cellular dehydration, altering ionic and osmotic pressures and impacting the accumulation of secondary metabolites in plants [90]. This stress can trigger the synthesis of secondary metabolites, serving as a defense mechanism against oxidative damage induced by ion accumulation at cellular and subcellular levels [91,92]. Salinity stress prompts the accumulation of organic and inorganic solutes in the root cell cytoplasm, lowering water potential and facilitating water uptake [93]. This process involves the synthesis and storage of compatible solutes (like proline) that function as osmotic adjustment agents, aiding plants in managing osmotic stress and safeguarding cell structures and macromolecules [94].

Salt-treated echinacea species, particularly *E. purpurea*, exhibited an increased capacity to exclude Na^+^ ions and showed enhanced antioxidant activity of enzymes such as ascorbate peroxidase (APX) and superoxide dismutase (SOD) [3]. However, catalase (CAT) activity decreased, and there was no significant change in the activity of glutathione reductase (GR) [95], as indicated in Figure 6. In some cases, the application of exogenous compatible solutes led to increased tolerance in plants subjected to salinity. For instance, plants treated with exogenous glycine betaine showed increased antioxidant capacity compared to control (untreated) plants [96]. Furthermore, inorganic ion accumulation is a strategy employed by salt-tolerant plants, including halophytes, to reduce osmotic potential. Ions such as Na^+^ and Cl^−^ are primarily stored in vacuoles and are utilized for osmotic adjustment in plant cells [89]. This accumulation of inorganic ions is an energy-efficient strategy compared to synthesizing organic substances [97].

Increased proline content has been linked to salt tolerance in *E. angustifolia* and *E. purpurea* under high NaCl concentration (60 mM NaCl) [3,16,92]. Salt stress can suppress the germination of *E. purpurea* seeds, leading to increased concentrations of the osmotic-adjustment substances such as proline and soluble sugar, higher activities of peroxidase (POD) and SOD, and the accumulation of Na^+^ ions along with a decrease in the K^+^/Na^+^ ratio [2,92] (Table 15).

## 6. Elicitors of Secondary Metabolites in Echinacea Species

Global warming has led to an increase in global temperatures, and this temperature rise is expected to have a significant influence on the production of secondary metabolites in plants [100]. Both high and low temperatures are considered abiotic stresses for plants, and they trigger specific responses affecting secondary metabolite biosynthesis [101]. High temperatures tend to promote the production of alkaloids such as hydroxycamptothecin in *E. purpurea* and *E. angustifolia* exposed to heat shock [102]. Production of phenylamides, known for their capacity to scavenge ROS, was stimulated in *E. purpurea* upon heat shock [103].

Cold temperatures may have a positive impact on the production of phenolic compounds in plants, and these compounds are subsequently modified and incorporated into the plant cell wall as lignin or suberin [104]. Echinacea species that are adapted to cold environments (e.g., *E. angustifolia*) tend to synthesize high levels of chlorogenic acid [105]. Furthermore, low temperatures favored the biosynthesis of ginsenosides in the root hairs of *E. angustifolia* and *E. purpurea* [106]. Biosynthesis of anthocyanin increased significantly in *E. pallida* plants exposed to low temperatures [107]. These findings underscore the dynamic relationship between temperature fluctuations and the production of secondary metabolites in echinacea species, indicating such temperature extremes can be employed as an effective tool to enhance secondary metabolite production in large-scale farming systems.

Growing latitude may also lead to increases in phenolic and alkylamide contents of *E. purpurea* and *E. angustifolia* [108]. Analysis of root extracts indicated that echinacea species grown at higher latitudes accumulated significantly higher amounts of total alkylamides and echinacoside, and the latter is a prominent phenolic marker used for grading the quality of *E. purpurea* root extracts [109].

The phenolic and antioxidant levels tend to increase in *E. angustifolia* plants that grow in a continental climate, characterized by significant day-to-night temperature fluctuations and long daylight hours during summer [110]. The lengthening of the photoperiod, along with milder temperatures that do not involve chilling, can influence how plants distribute photoassimilates between source and sink organs. Longer daylight periods promote the export of photoassimilates from photosynthesizing leaves to other organs that serve as storage sites [111].

Plants that survive harsh winter conditions commonly undergo a process known as cold acclimation, which involves accumulating complex carbohydrates and secondary metabolites to withstand and prevent injury caused by freezing temperatures [112]. It has been proposed that *E. angustifolia* and *E. pallida* may employ a similar mechanism to survive under freezing stress [113]. Phenolic compounds and flavonol glycosides, functioning as anti-ice nucleators, initiate a supercooling mechanism to hinder the formation of large ice crystals in the roots [34].

## 7. Factors Influencing the Quality of Echinacea Roots for Commercial Use

Supplying consistently high-quality echinacea roots for commercial purposes presents challenges. The medicinal and phytochemical components in echinacea roots can vary significantly from one harvest to another due to various agronomic/environmental factors. These factors encompass seed stock quality, soil type, planting time, seed viability, moisture levels, temperature fluctuations, fertilization methods, biotic/abiotic stresses, mycorrhizal colonization, pest control measures, harvest timing, and post-harvest handling procedures [15,27]. Biomass of root and flower heads in echinacea species can be significantly influenced by the growing conditions [114].

Some agricultural techniques boost the buildup of phenolic compounds, such as chicoric acid, caftaric acid, and chlorogenic acid, in both roots and shoots of *E. purpurea* [115]. These factors include foliar application of plant stress mediators such as salicylic acid and its derivatives, as well as the use of a metal elicitor such as titanium (IV) ascorbate [116]. Furthermore, plant density seems to be a key factor when it comes to biosynthesis of secondary metabolites in field-grown echinacea plants. For example, cultivating *E. purpurea* at high density (more than 15 plants per square meter) in field conditions has been associated with a reduction in chicoric acid accumulation [117].

Among the various factors affecting echinacea cultivation, fertilization practices, and harvest timing appear to exert the most significant influence on the phytochemical composition of the plants. Notably, differences in fertilization, particularly nitrogen and potassium rates, have been found to alter alkylamide contents in *E. purpurea* [3] and *E. angustifolia* [118]. The high rate of nitrogen application resulted in increased plant yield and essential oil production in *E. purpurea* [119]. The composition and levels of phenolic acid compounds can be influenced by soil type (sandy vs. loamy) and fertilization techniques [27]. Additionally, the timing of harvest can impact alkamide levels, with samples collected in different months exhibiting significant differences [120]. Root biomass in echinacea plants tends to increase significantly with plant age [121].

## 8. Hydroponic Cultivation

The accumulation of secondary metabolites in echinacea plants can also be influenced by field cultivation as opposed to hydroponic production [3,122]. Hydroponic production of echinacea has gained increasing attention due to several factors adversely affecting the traditional soil-based cultivation system [3,16,123]. The soil-related issues include inconsistencies in crop growth, seed dormancy, the risk of adulteration with misidentified species, the presence of soil-borne pathogens, and concerns related to chemical contaminants in soil, such as heavy metals [124].

The interest in hydroponic culture for cultivating medicinal plants, including echinacea, has been steadily increasing [125]. Hydroponics offers advantages by providing precise control over the nutrient solution, which allows for the regulation of secondary metabolism in *E. purpurea* [3,16,21].

Hydroponically grown *E. angustifolia* plants can exhibit higher levels of certain secondary metabolites compared to those cultivated in traditional (field) conditions [120]. For instance, the concentration of echinacoside, a major caffeoyl conjugate found in *E. angustifolia* root, was more than 2-fold the quality standards threshold when the plants were grown hydroponically [121]. *E. purpurea* plants grown hydroponically [3,16,21] produced more chicoric acid compared to field-grown plants harvested after 3 years [117]. Echinacea plants cultivated hydroponically, particularly *E. purpurea* and *E. angustifolia*, had higher concentrations of caftaric acid, cynarin, echinacoside, and chicoric acid compared to those grown in the wild or in managed field conditions [104]. The increased accumulation of these phytochemicals in hydroponic cultivation could be due to the abundant nutrient supply in the growth medium or the preservation and selective harvesting of fine roots known for their rich content of bioactive compounds [3,16,21]. Hydroponic systems effectively retain valuable fine roots containing high concentrations of chicoric acid, contrasting with traditional field harvesting where a substantial portion of these roots is lost during washing or is left in the ground. In summary, hydroponic cultivation offers superior efficiency over field methods, ensuring greater yields of secondary metabolites and faster growth cycles.

## 9. Knowledge Gaps and Future Research Directions

Most echinacea supplements utilize extracts from the roots or flower heads, occasionally incorporating leaves, from *E. purpurea*, *E. angustifolia*, or *E. pallida*. However, each part of the echinacea plant harbors distinct secondary metabolites, suggesting diverse medicinal applications. Exploring the untapped potential of other plant parts and lesser-known echinacea species is a promising avenue for both research and industry. Limited information exists on the medicinal and agricultural attributes of rare echinacea species that may offer novel compounds absent in commercially utilized varieties. In vitro culturing systems present an opportunity to study and cultivate these rare species without endangering their natural habitats.

In the future, a blend of tissue culture and hydroponic methods is poised to set a new benchmark for echinacea production, enhancing both yield and phytochemical quality. Biotechnological innovations offer diverse avenues for optimizing secondary metabolite synthesis in echinacea plants, paving the way for a wider array of echinacea products to meet the expanding market demands. Further exploration is necessary to refine cultivation techniques and select high-yielding genotypes for hydroponic systems, particularly in achieving optimal accumulation of key secondary metabolites like CADs. Challenges such as low seed germination rates and lengthy maturation times persist, especially in field settings, underscoring the need for standardized product formulations in future endeavors.

## Figures and Tables

**Figure 1 plants-13-01235-f001:**
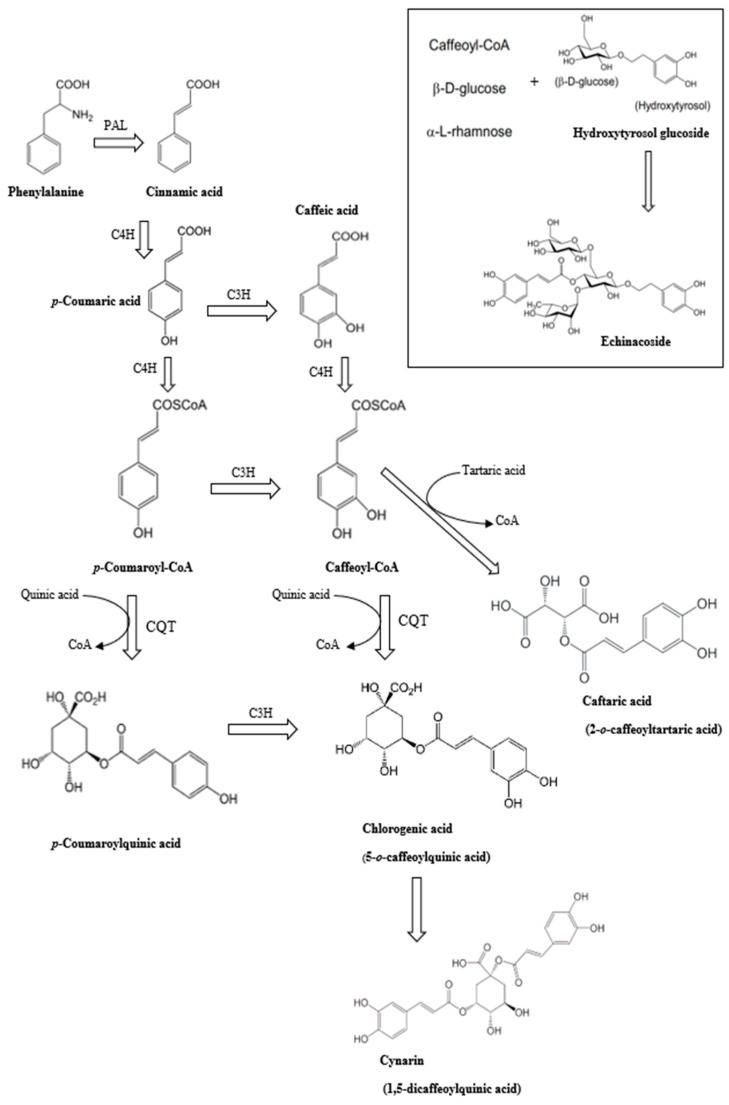
Possible biosynthetic route for caffeic acid derivates (CADs) via phenylpropanoid pathway in echinacea species. PAL, phenylalanine ammonia-lyase; C4H, cinnamate 4-hydroxylase; C3H, *p*-coumarate-3-hydroxylase; 4CL, 4-(hydroxyl)cinnamoyl CoA ligase; CQT, caffeoyl-CoA/quinic acid caffeoyl transferase; HTT, hydroxycinnamoyl-CoA/tartaric acid hydroxycinnamoyl transferase; UGT UDP, glucose transferase.

**Figure 2 plants-13-01235-f002:**
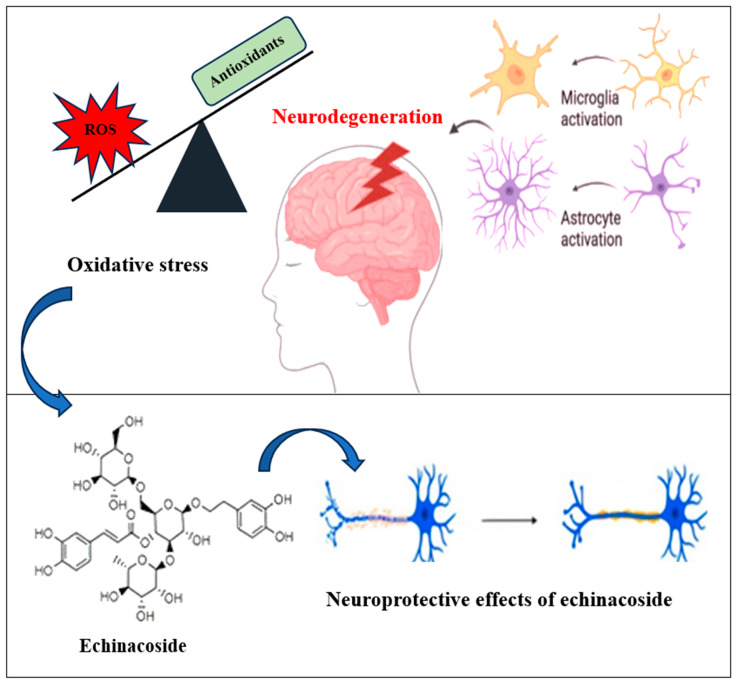
The mechanism of echinacoside involved in the neuroprotective effect.

**Figure 3 plants-13-01235-f003:**
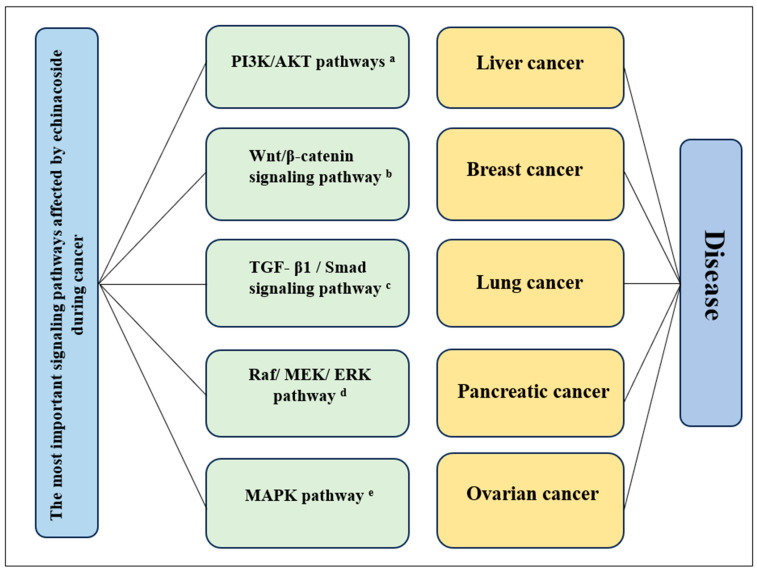
Echinacoside activity in different cancer types. ^a^ PI3K/AKT pathways: The phosphoinositide-3-kinase–protein kinase B/Akt (PI3K-PKB/Akt) pathway. ^b^ Wnt/β-catenin signaling pathway: Wingless/Integrated. ^c^ TGF-β1/Smad signaling pathway: transforming growth factor-β (TGF-β) pathway inhibitor. ^d^ Raf/MEK/ERK pathway: Rapidly accelerated fibrosarcoma (Raf)/mitogen-activated protein kinase (MEK)/extracellular signal-regulated kinase (ERK). ^e^ MAPK pathway: MAPK (mitogen-activated protein kinase) signaling pathways.

**Figure 4 plants-13-01235-f004:**
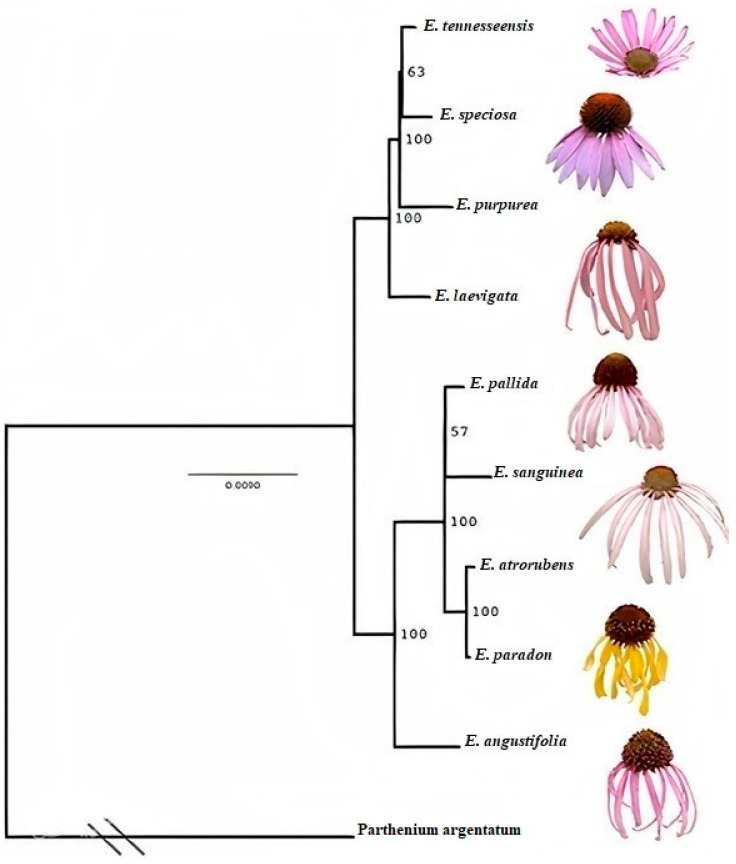
The maximum likelihood tree of echinacea reconstructed using chloroplast genomes. Numbers on branch nodes are bootstrap values. The branch connecting the outgroup *Parthenium argentatum* and nine echinacea species is collapsed [69].

**Figure 5 plants-13-01235-f005:**
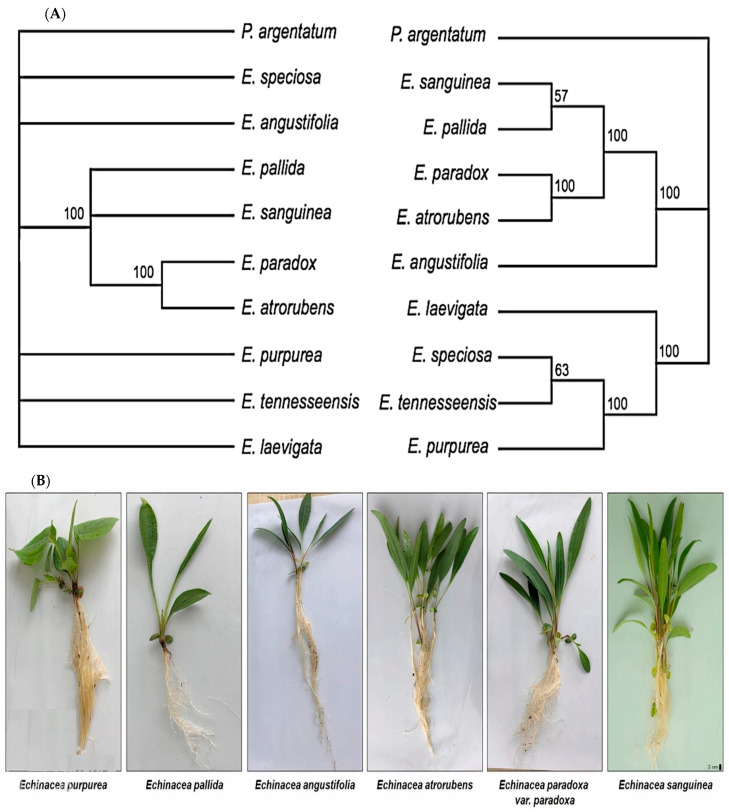
(**A**) The maximum likelihood tree reconstructed using maK + rbcL (left) and chloroplast genomes (right). Numbers are bootstrap values, and branches with bootstrap values < 50% are collapsed [70]. (**B**) Morphology of six echinacea species [71].

**Figure 6 plants-13-01235-f006:**
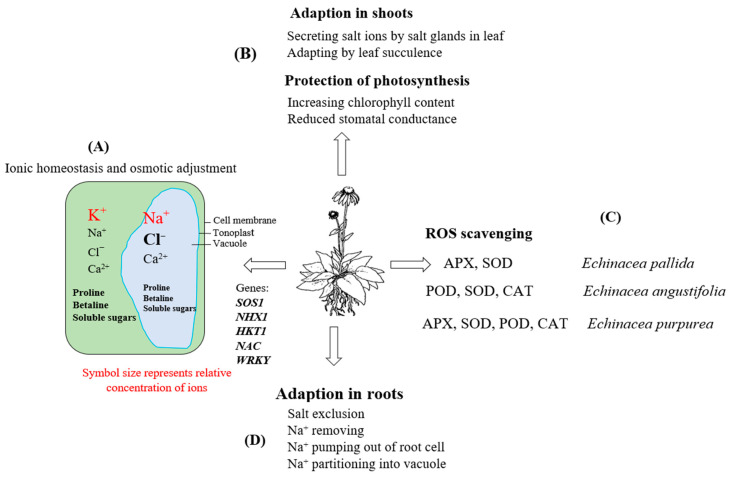
Graphical summary of the plant adaptation mechanisms under salinity conditions. (**A**) Adaptation through ion homeostasis and osmotic adjustment. (**B**) Adaptation through salt secretion, leaf succulence, photosynthesis protection, and reduction of water loss in shoots. (**C**) Adaptation through scavenging ROS. (**D**) Adaptation through salt exclusion via pumping sodium ions out of root cells. SOS1: Salt Overly Sensitive 1; NHX1: vacuolar Na^+^/H^+^ antiporter; HKT1: high-affinity K^+^ transporters; WRKY and NAC: transcription factors; ROS: reactive oxygen species; SOD: superoxide dismutase; CAT: catalase; APX: ascorbate peroxidase; POD: peroxidase.

**Table 1 plants-13-01235-t001:** Localization of bioactive compounds in echinacea species.

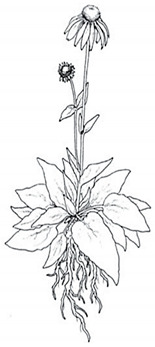	*E. purpurea*	*E. angustifolia*	*E. pallida*	*E. paradoxa*
	Flower head			
	**Caffeic acid derivatives**	Caffeic acid derivatives	Caffeic acid derivatives	**Alkylamides**
	*Alkylamides*	Alkylamides	**Alkylamides**	*Echinacoside*
		**Echinacoside**	*Echinacoside*	
	Stem			
	Alkylamides	**Alkylamides**	Alkylamides	**Alkylamides**
	*Chicoric acid*	Chicoric acid	*Caftaric acid*	Caftaric acid
	Leaves			
	Chicoric acid	Chicoric acid	**Chicoric acid**	Alkylamides
	**Caftaric acid**	*Caftaric acid*	*Alkylamides*	*Echinacoside*
	Cynarin	Cynarin		
	Root			
	**Caffeic acid** **derivatives** **Cynarin** **Echinacoside**	**Caffeic acid** **derivatives** Cynarin	*Echinacoside* Alkylamides	**Echinacoside** *Alkylamides*

Note: The compounds highlighted in bold are more abundant in various parts of *Echinacea* species when compared to the other compounds. Italics indicate compounds found in trace amounts [15].

**Table 9 plants-13-01235-t009:** The most common polysaccharides in *E. purpurea*.

Polysaccharides	Molecular Mass (kDa)	Reference
Inulin-type fructan	6	[31]
Xyloglucan	79.5	[19]
Acid rhamnoarabinogalactan	45	[20]
Acidic arabinogalactan	70	[47]
Arabinogalactan-protein	1200	[30]
4-*o*-methylglucuronoarabinoxylan	35	[7]
Heterogeneous polysaccharides	10–50	[18]

**Table 10 plants-13-01235-t010:** Neuroprotective properties of echinacoside (ECH).

Disease	Dosage/Concentration	Study Models	Key Findings	References
Parkinson’s disease	5 and 20 mg/kg per day for 15 days	Animal model	ECH improves the behavioral and neurochemical outcomes in the MPTP mice model of Parkinson’s disease and inhibits caspase-3 and caspase-8 activation in cerebellar granule neurons	[83]
Parkinson’s disease	5, 10, 20 mg/L	In vitro	In neuronal cells, ECH activates the Trk-extracellular signal-regulated kinase (ERK) pathway, leading to the inhibition of cytochrome c release and caspase-3 activation induced by subsequent rotenone exposure.	[79]
Parkinson’s disease	0.1, 0.5, 1, 5, 10 μM	In vitro	The potential mechanism of ECH in countering 6-OHDA-induced neurotoxicity may involve decreased ROS production, leading to the attenuation of mitochondrial dysfunction and inflammatory response	[39]
Parkinson’s disease	20 mg/kg per day for 15 days 10, 20, 40 μg/mL	Animal model/in vitro	In Parkinson’s disease, the ROS/ATF3/CHOP pathway is highly important in the protective effects of ECH against MPTP-induced apoptosis.	[28]
Epilepsy	5, 10, 50 mg/kg per day	Animal model	In a kainic acid rat model, ECH exerts its antiepileptic and neuroprotective effects by suppressing inflammatory response and activating the Akt/GSK3β signaling	[47]
Depressive disorders	20, 30, 40 mg/kg	Animal model	ECH could provide antidepressant-like effects in mice via the activation of the AMPAR-Akt/ERK-mTOR pathway in the hippocampus	[38]
Alzheimer’s disease	2.5 and 5.0 mg/kg per day for 15 days 32 and 64 μM	Animal model/in vitro	ECH alleviated cognitive dysfunction resulting from Aβ 1–42 by inhibiting amyloid oligomerization, preventing amyloid deposition, and mitigating cortical cholinergic neuronal impairment through the reduction of amyloid neurotoxicity	[90]
Spinal cord injury	20 mg/kg per day for 35 days	Animal model/in vitro	By suppressing the NLRP3 inflammasome-related signaling pathway, ECH has the potential to enhance motor function recovery in rats with a spinal cord injury	[36]

**Table 11 plants-13-01235-t011:** Anticancer properties of echinacoside (ECH).

Disease	Dosage/Concentration	Study Models	Key Findings	References
Colorectal cancer	20, 40, 80 mg/kg per day	Animal model	ECH exhibits oral antimetastatic efficacy by facilitating butyrate-producing gut bacteria, which downregulates PI3K/AKT signaling and epithelial-mesenchymal transition.	[47]
Colorectal cancer	60, 80, 150 μM	in vitro	ECH triggers cell cycle arrest and apoptosis in SW480 cancer cells by causing oxidative DNA damage.	[45]
Breast cancer	20, 50, 75, 100 μM 10 mg/kg	Animal model/in vitro	Treatment with ECH resulted in a significant decrease in tumor growth, correlating with the inhibition of Wnt/β-catenin signaling	[12]
Breast cancer	30 and 20 mg/mL for 6, 12, 24 h	in vitro	The ethyl acetate extract resulted in cell cycle arrest at the G1 phase and induced apoptosis through caspase activation.	[42]
Breast cancer	5, 10, 20, 40 μg/mL for 1–6 days	in vitro	ECH inhibited cell proliferation, invasion, and migration, and promoted the apoptosis of breast cancer cells by downregulating the expression of miR-4306 and miR-4508.	[44]
Liver cancer	5 mg/kg per day for 4 weeks	Animal model/in vitro	ECH and AMPG exhibited superior effects on hepatocellular carcinoma (HCC) cells compared to free ECH, illustrating its potential for HCC chemotherapy due to the nontoxic nature of AMPG and high drug-loading capacity.	[39]
Liver cancer	20, 50, 100 μg/mL 20 and 50 mg/kg	Animal model/in vitro	The antitumor activity of ECH was observed through the downregulation of TREM2 expression and inhibition of the PI3K/AKT signaling pathway.	[40]
Liver cancer	5, 10, 20 mg/mL	in vitro	ECH promoted the activation of the TGF-β1/Smad signaling pathway and increased the expression levels of Bax/Bcl-2 in liver cancer cells. Moreover, ECH could trigger the release of mitochondrial Cyto C.	[52]

**Table 12 plants-13-01235-t012:** Hepatoprotective properties of echinacoside (ECH).

Disease	Dosage/Concentration	Study Models	Key Findings	References
Chemical-induced liver injury	10, 30, 60 mg/kg per day	Animal model/in vitro	ECH protects against ethanol-induced liver injuries by alleviating oxidative stress and cell apoptosis via increasing the activity of Nrf2.	[33]
Chemical-induced liver injury	60 mg/kg per day	Animal model	ECH exhibits both anti-apoptotic and anti-inflammatory effects, evident by its notable suppression of hepatocyte apoptosis and a significant reduction in inflammatory markers.	[52]
Chemical-induced liver injury	20 mg/kg per day	Animal model	The hepatoprotective effect of ECH was achieved through the inhibition of inflammatory factor release by the TLR4/NF-κB signaling pathway.	[35]
Chemical-induced liver injury	50 and 100 μM 100 mg/kg per day	Animal model/in vitro	ECH exerts protective effects against ethanol-induced liver injuries by attenuating oxidative stress and hepatic steatosis by modulating the SREBP-1c/FASN pathway via PPAR-α	[56]
Chemical-induced liver injury	50 mg/kg per day	Animal model	ECH administration significantly reduced serum ALT and AST levels, hepatic MDA content, and ROS production. Additionally, it restored hepatic SOD activity and GSH content.	[51]
Chemical-induced liver injury	25 and 100 mg/kg	in vitro	ECH inhibited the elevation of sAST and sALT levels in D-GalN/LPS-treated mice and thus decreased the sensitivity of hepatocytes to TNF-α.	[55]
Drug-induced liver injury	25, 50, 100 mg/kg per day	Animal model	ECH exhibited a substantial protective effect against acetaminophen-induced hepatotoxicity by attenuating oxidative stress, suppressing the expression of proinflammatory cytokines, and reducing cytochrome P450 2E1 protein expression.	[53]
Viral hepatitis	1,10, 25, 50, 100 mg/L	Animal model/in vitro	ECH exhibited strong inhibitory effects on HBV replication and antigen expression.	[54]
Hepatic fibrosis	500, 250, 125 µg/mL	in vitro	ECH blocked the TGF-β1/Smad signaling pathways and inhibited the activation of hepatic stellate cells.	[50]

Nrf2: nuclear factor erythroid 2; TLR4/NF-κB: toll-like receptor 4/nuclear factor kappa B; SREBP-1c/FASN: sterol regulatory element-binding protein-1c/fatty acid synthase; ALT: alanine aminotransferase; AST: aspartate aminotransferase; MDA: malondialdehyde; SOD: superoxide dismutase; GSH: glutathione; sAST: serum aspartate aminotransferase; sALT: serum alanine aminotransferase; D-GalN/LPS: D-galactosamine/lipopolysaccharide; TNF-α: tumor necrosis factor-alpha; HBV: hepatitis B virus; TGF-β1: transforming growth factor beta 1.

**Table 13 plants-13-01235-t013:** Number (bottom triangle) and percentage (top triangle) of differences among nine echinacea chloroplast genomes [72].

	*paradox*	*atrorubens*	*sanguinea*	*pallida*	*angustifolia*	*tennessensis*	*leavigata*	*seciosa*	*purpurea*
*paradox*		0.12%	0.23%	0.18%	0.44%	0.52%	0.51%	0.50%	0.56%
*atrorubens*	181		0.20%	0.18%	0.48%	0.55%	0.55%	0.55%	0.60%
*sanguinea*	345	308		0.16%	0.45%	0.54%	0.53%	0.54%	0.60%
*pallida*	273	276	247		0.41%	0.50%	0.50%	0.50%	0.55%
*angustifolia*	672	727	685	629		0.47%	0.45%	0.45%	0.53%
*tennessensis*	787	837	827	765	711		0.29%	0.20%	0.31%
*leavigata*	772	835	813	764	677	445		0.24%	0.31%
*seciosa*	768	830	827	767	689	309	365		0.23%
*purpurea*	849	910	908	842	811	469	478	350	

**Table 14 plants-13-01235-t014:** The 25 most-divergent non-coding regions among nine echinacea species [73].

Genes	Length (bp)	Variable Sites	Indels	Percentage of Identical Sites (%)
*ccsA→trnL-UAG*	138	2	3	81.9
*psbI→trnS-GCU*	144	4	5	86.8
*5 S rRNA→trnRACG*	312	0	2	86.9
*atpF→atpA*	72	0	2	88.9
*rpl32→ndhF*	904	4	7	88.9
*trnT-UGU→trnLUAA*	603	5	8	90.9
*petN→psbM*	539	3	4	90.9
*rps4→trnT-UGU*	392	3	3	91.6
*petD→rpoA*	205	3	3	91.7
*ndhI→ndhG*	388	3	1	92.5
*trnT-GGU→psbD*	1270	11	8	92.9
*ndhD→ccsA*	234	2	4	93.2
*trnH-GUG→psbA*	385	8	4	93.2
*trnK-UUU→matK*	304	1	3	93.4
*psbC→trnS-UGA*	246	1	3	93.6
*ndhC→trnV-UAC*	998	9	7	93.9
*ycf3→trnS-GCU*	910	8	4	94.0
*trnK-UUU→rps16*	783	2	5	94.1
*trnR-UCU→trnGUCC*	221	5	2	94.6
*rps8→rpl14*	203	1	3	94.6
*psaA→ycf3*	747	6	5	94.9
*psaI→ycf4*	396	0	2	94.9
*rpoC2→rps2*	259	0	2	95.0
*rbcL→accD*	580	3	2	95.0
*rps2→atpI*	233	1	1	95.3

**Table 15 plants-13-01235-t015:** Quantitative variations in the accumulation of some secondary metabolites in different echinacea species subjected to salinity conditions.

NaCl	Plant Species	Compounds	Plant Organ	Response	Reference
50 µM	*E. purpurea*	Total phenols, total flavonoids	Leaf	+	[98]
100 mM (+Si)	*E. purpurea*	Chlorogenic acid, caftaric acid, total phenols	Root	+	[68]
30 mM	*E. purpurea*	Echinacoside, caffeic acid, chicoric acid, chlorogenic acid, cynarine	Root	+	[3]
30 mM	*E. purpurea*	Total phenols, total flavonoids, antioxidant activity	Root	+	[92]
50 mM	*E. angustifolia*	Chicoric acid, chlorogenic acid	Root	+	[90]
50 µM	*E. angustifolia*	Echinacoside	Root	−	[99]
50 mM	*E. purpurea*	Total flavonoids	Root	+	[15]
75 mM	*E. purpurea*	Caftaric acid, cynarine	Root	+	[95]
75 mM	*E. pallida*	Caftaric acid, echinacoside	Root	+	[95]
75 mM	*E. angustifolia*	Chicoric acid	Root	+	[95]
75 mM	*E. angustifolia*	Alkamides	Root	−	[95]
100 mM	*E. purpurea*	Caftaric acid	Leaf	−	[19]
50 mM	*E. purpurea*	Total flavonoids	Leaf	−	[15]
60 mM	*E. purpurea*	Total phenols, total flavonoids, antioxidant activity	Leaf	+	[23]
12 dS m^−1^	*E. purpurea*	Total phenols, polysaccharides	Leaf	+	[91]
150 mM	*E. purpurea*	Antioxidant activity, soluble sugars	Root	−	[93]
6 dS m^−1^	*E. purpurea*	Germination rate	Seed	−	[75]
50 mM	*E. angustifolia*	Morphological properties	Root	−	[90]
60 mM	*E. purpurea*	Chicoric acid, echinacoside, caftaric acid	Root	−	[3,16]

## Data Availability

Data and original figures are available upon request to the corresponding author.

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
