# Peer review of "Echinacea: Bioactive Compounds and Agronomy"

_plants, 2024, doi:10.3390/plants13091235_

Round 1
Reviewer 1 Report
Comments and Suggestions for Authors
In this manuscript (plants-2969408) entitled "Echinacea: Bioactive Compounds and Agronomy" submitted to Plants, Fatemeh Ahmadi and colleagues have provided a comprehensive analysis of the existing knowledge on recent advances in understanding the physiology, secondary metabolites, agronomy, and ecology of echinacea plants, focusing on E. purpurea, E. angustifolia, and E. pallida. Authors highlight challenges in echinacea research and provide insights into diverse approaches to boost the biosynthesis of secondary metabolites of interest in echinacea plants and optimise their large-scale farming. This review is interesting and well-written, but the current version of this manuscript needs to be revised for publication.
Major points:
1. For the Table 1, authors should consider to include citations in the revision.
2.For the Figures, the sentence “Figure drawn by F. Ahmadi” should be removed from the revised legend.
3.For the Figure 2 and 3, authors should consider to redraw these two figures. The present figure looks like modified from existing Figures and are unsuitable for publication.
4, For the Figure 4, Plant species names in the text should not be written in strange fonts, please revise.
5, For the Figure 5B, scale bar should be included in the revised pictures.
Minor points:
1. Plant species names should be in italics. Authors should check all plant species names employed in the manuscript.
2. Authors need to standardize references according to the Plants template.
Author Response
Journal: Plants
Manuscript title: Echinacea: Bioactive Compounds and Agronomy
Manuscript number: 2969408
Authors: Fatemeh Ahmadi, Khalil Kariman, Milad Mousavi, Zed Rengel
We are very thankful to the editorial board and reviewers for their comments on the manuscript. All comments were addressed in the revised manuscript.
Reviewer #1:
In this manuscript (plants-2969408) entitled "Echinacea: Bioactive Compounds and Agronomy" submitted to Plants, Fatemeh Ahmadi, and colleagues have provided a comprehensive analysis of the existing knowledge on recent advances in understanding the physiology, secondary metabolites, agronomy, and ecology of echinacea plants, focusing on E. purpurea, E. angustifolia, and E. pallida. The authors highlight challenges in echinacea research and provide insights into diverse approaches to boost the biosynthesis of secondary metabolites of interest in echinacea plants and optimize their large-scale farming. This review is interesting and well-written, but the current version of this manuscript needs to be revised for publication.
Major points:
- For Table 1, authors should consider to include citations in the revision.
Corrected. References were added to the reference list. Page 3, line 88, and page 36 lines 1071-1077.
- For the Figures, the sentence “Figure drawn by F. Ahmadi” should be removed from the revised legend.
Corrected. The sentence was removed from all Figure's captions.
- For Figures 2 and 3, the authors should consider redrawing these two figures. The present figure looks like modified from existing Figures and is unsuitable for publication.
Corrected. Figures 2 and 3 were redrawn. Pages 13 and 16.
- For Figure 4, Plant species names in the text should not be written in strange fonts, please revise.
Corrected. The font of plant species was revised in Figure 4. Page 21.
- For Figure 5B, the scale bar should be included in the revised pictures.
Corrected. The scale bar was included in the revised Figure (Figure 5).
Minor points:
- Plant species names should be in italics. Authors should check all plant species names employed in the manuscript.
Corrected. All species names were checked for italics.
- Authors need to standardize references according to the Plants template.
Corrected. References were corrected based on the journal format. Pages 32-38.
Reviewer 2 Report
Comments and Suggestions for Authors
Although there are numerous studies and even reviews in the specialty literature, the approach is unique and very complex by comparing the 3 species, respectively the parts of the respective plant.
Author Response
Reviewer #2:
Although there are numerous studies and even reviews in the specialty literature, the approach is unique and very complex by comparing the 3 species, respectively the parts of the respective plant.
The authors are thankful for your opinion.
Reviewer 3 Report
Comments and Suggestions for Authors
This review provides a comprehensive analysis of recent advances in the physiology,
secondary metabolites, agronomy, and ecology of Echinacea plants, and a detailed
discussion of pharmacological activities. The authors provide valuable insights into
the biosynthesis of secondary metabolites and various methods to optimize their
large-scale cultivation of Echinacea plants. It is of interest to the readers.
Page 3 line 99 the polysaccharides enhanced
Change “the polysaccharides enhanced”into “the polysaccharides enhance”
Page 4 line 109 standardize echinacea products based on their content
Change “standardize echinacea products based on their content” into “standardized
echinacea products based on their content”
Page 16 line 955 Ahmadi F, Samadi A, Sepehr E, Rahimi A, Shabala S. 2022a.
Change “Ahmadi F, Samadi A, Sepehr E, Rahimi A, Shabala S. 2022a. ”into “Ahmadi
F, Samadi A, Sepehr E, Rahimi A, Shabala S. 2022. ”
Page 18 line 1050 47. Liu J, Tang N, Liu N, Lei P, Wang F.
Change “47. Liu J, Tang N, Liu N, Lei P, Wang F.” into “Liu J, Tang N, Liu N, Lei P,
Wang F. ”
Page 20 line 1142 Ahmadi F, Samadi A, Sepehr E, Rahimi A, Shabala S. 2022b.
Change “Ahmadi F, Samadi A, Sepehr E, Rahimi A, Shabala S. 2022b. ” into
“Ahmadi F, Samadi A, Sepehr E, Rahimi A, Shabala S. 2022. ”
Please carefully check and verify all references, and revise them in accordance with
the requirements of the journal.

The quality of English is very good.
Author Response
Reviewer #3:
This review provides a comprehensive analysis of recent advances in the physiology, secondary metabolites, agronomy, and ecology of Echinacea plants, and a detailed discussion of pharmacological activities. The authors provide valuable insights into the biosynthesis of secondary metabolites and various methods to optimize their large-scale cultivation of Echinacea plants. It is of interest to the readers.
- Page 3 line 99 the polysaccharides enhanced: Change “the polysaccharides enhanced” into “the polysaccharides enhance”
Corrected. Page 3 line 94.
- Page 4 line 109 Standardize echinacea products based on their content: Change “standardize echinacea products based on their content” into “standardized echinacea products based on their content”
Corrected. Page 3 line 103.
- Page 16 line 955 Ahmadi F, Samadi A, Sepehr E, Rahimi A, Shabala S. 2022a. Change “Ahmadi F, Samadi A, Sepehr E, Rahimi A, Shabala S. 2022a. ”into “Ahmadi F, Samadi A, Sepehr E, Rahimi A, Shabala S. 2022. ”
Corrected. Page 33 line 830.
- Page 18 line 1050 47. Liu J, Tang N, Liu N, Lei P, Wang F: Change “47. Liu J, Tang N, Liu N, Lei P, Wang F.” into “Liu J, Tang N, Liu N, Lei P, Wang F. ”
Corrected. Page 34 line 886.
- Page 20 line 1142 Ahmadi F, Samadi A, Sepehr E, Rahimi A, Shabala S. 2022b. Change “Ahmadi F, Samadi A, Sepehr E, Rahimi A, Shabala S. 2022b. ” into “Ahmadi F, Samadi A, Sepehr E, Rahimi A, Shabala S. 2022. ”
Corrected. Page 36 line 979.
- Please carefully check and verify all references, and revise them according to the requirements of the journal.
Corrected. References were corrected based on the journal format. Pages 32-38.